# Selenoproteins in Health

**DOI:** 10.3390/molecules29010136

**Published:** 2023-12-25

**Authors:** Ziqi Qi, Alex Duan, Ken Ng

**Affiliations:** 1School of Agriculture, Food and Ecosystem Sciences, Faculty of Science, The University of Melbourne, Parkville, VIC 3010, Australia; ziqiq1@student.unimelb.edu.au; 2Melbourne TrACEES Platform, School of Chemistry, Faculty of Science, The University of Melbourne, Parkville, VIC 3010, Australia; duanx@unimelb.edu.au

**Keywords:** selenium, selenoprotein, bioaccessibility, bioactivity, food resources

## Abstract

Selenium (Se) is a naturally occurring essential micronutrient that is required for human health. The existing form of Se includes inorganic and organic. In contrast to the inorganic Se, which has low bioavailability and high cytotoxicity, organic Se exhibits higher bioavailability, lower toxicity, and has a more diverse composition and structure. This review presents the nutritional benefits of Se by listing and linking selenoprotein (SeP) functions to evidence of health benefits. The research status of SeP from foods in recent years is introduced systematically, particularly the sources, biochemical transformation and speciation, and the bioactivities. These aspects are elaborated with references for further research and utilization of organic Se compounds in the field of health.

## 1. Introduction

Selenium (Se) is a naturally occurring essential micronutrient that plays an important role in human health. In the 1970s, Rotruck conducted a seminal investigation into the role of Se and determined that it is an integral component of the antioxidant enzyme known as glutathione peroxidase. In cases of Se deficiency, the full functionality of this enzyme is impaired, underscoring the pivotal role of See in maintaining overall bodily health [1]. This finding established Se’s significance in bolstering cellular antioxidant defenses, as glutathione peroxidase serves as a primary antioxidant enzyme within cells. In 1973, the World Health Organization (WHO) officially recognized Se as an essential trace element critical to the vital processes of both humans and animals. Subsequently, extensive research endeavors were initiated to elucidate the intricate connections between Se-containing compounds and human well-being, as well as to determine the dietary requisites for Se and Se levels in various food sources.

The WHO and the Food and Agriculture Organization (FAO) eventually established the recommended dietary allowance (RDA) for Se at 55 μg/day for adult men and women above the age of 19 [2]. Pregnant or lactating women are advised to consume 60 μg/day and 70 μg/day, respectively. The Tolerable Upper Intake Level (UL) for Se for all populations is 400 μg/day [3]. Daily Se intake varies according to the geographical origin of foods and the dietary preferences of local populations. For instance, in Australia, the daily Se intake averages around 87 μg/day for men and 57 μg/day for women [4]. In the United States, the figures are notably higher, with 134 μg/day for men and 93 μg/day for women [5]. In Europe, the dietary Se intake typically falls within a range of 40 to 60 μg/day across most regions [6]. However, Se deficiency, now recognized as a significant detriment to health, continues to be a global challenge. This predicament largely stems from the inadequate Se content in numerous food sources and staples cultivated in Se-deficient soil, as well as livestock consuming Se-deficient feed [7,8]. Selenium deficiency is linked to several health conditions, including cardiovascular disease, liver disease, thyroid autoimmune disease, cognitive dementia, cancer, Type II diabetes, and specific ailments such as Kashin-Beck disease (KBD) and osteoarthritis (OA). KBD, a skeletal developmental disorder, has been observed in regions with low selenium levels, such as parts of North Korea, China, and the Siberian region of Russia, and can be improved by supplementing with Se. OA, the most prevalent form of arthritis, results from a breakdown in cartilage due to imbalances in matrix metabolism. The onset of OA is often associated with oxidative stress, which plays a significant role in its development. Insufficient Se and disturbances in selenoproteins (SePs) have been linked to disruptions in the redox balance within cartilage, contributing to these conditions [9,10,11].

The Se content in plants is characterized by great diversity, which mainly depends on the Se level in the soil in which the plant grows and the ability of the plant species to assimilate Se. However, irregular Se contents have been measured in different regions of the world. Notably, the Se content of soils in most geographical areas ranges from 0.01 to 2 mg/kg, and Se-deficient areas are classified as those with less than 0.6 mg/kg [12,13]. Low Se areas include New Zealand, central Siberia, Denmark, Poland, and a belt from south-central to northeast China. Australia has both low- and high-Se soils. Selenium-deficient regions are known as southwest Western Australia, the south-eastern coast of Queensland, coastal and central regions of Victoria, a large part of Tasmania, as well as parts of NSW and South Australia. Seleniferous soils occur in parts of Cape York Peninsula and central Queensland [14]. Some geographical areas such as the Enshi region in Hubei Province, China, are high in Se, with more than 100 mg/kg [15]. High Se soils also occur in northern California and northwest India, at approximately 30 mg/kg and 4mg/kg, respectively [16,17]. Some factors that can affect selenium contents in the environment include soil type, higher rainfall (over 450 mm), heavy fertilizer application, and rapid pasture growth in spring. Granite or sandy acidic basalt are commonly low in trace minerals. High rainfall can leach minerals out of the topsoil layer, and the application of heavy fertilizer, particularly sulfur-fortified superphosphate or gypsum fertilizers, and rapid pasture growth can affect mineral accumulation in plants.

Chemically, Se naturally exists as both inorganic and organic forms. Inorganic Se species, such as selenite and selenate, are broadly distributed in soil and typically applied for growing agricultural products. The inorganic Se from soil can be incorporated into amino acids replacing the sulfur atom as selenoamino acids in plants, such as selenomethionine (SeMet), selenocysteine (SeCys) and methylselenocysteine (MeSeCys). Further processing of selenoamino acids produces SePs.

The health protectivity effect of Se does not only depend on its content obtained from foods but also on its chemical forms. Inorganic Se compounds can be used as Se sources in dietary supplements due to their cost-effectiveness, for example, when added to infant milk formula. However, it is still a challenge for safe and effectual supplementation considering inorganic Se’s low bioavailability, high cytotoxicity, and short retention time in the gastrointestinal tract. Numerous evidence indicates that organic Se from foods is more effective than inorganic Se in providing antioxidant protection to cells and tissues. By doing so, it improves immune function and immunomodulation, and provides anti-cancer and anti-inflammation effects [18,19,20,21,22]. Organic Se can be found in dietary suitable levels in some crops and vegetables (e.g., rice, buckwheat grain, mushroom, lettuce, potato, tea, etc.) cultivated in Se sufficient soil, and can also be found in some animals and animal products produced by animals with sufficient Se intakes (e.g., tuna, chicken, eggs, milk products, etc.). In addition, foods contain a variety of Se species, and the Se profiles of foods vary markedly.

Organic Se exists in various forms with distinct physiological effects within the human body. SeMet, found in grains and legumes, incorporates into proteins, influencing synthesis and acting as a major storage form. SeCys, an essential component of selenoproteins, plays crucial roles in antioxidant defense and thyroid hormone metabolism. MeSeCys, present in garlic and onions, is associated with potential anticancer properties. Inorganic forms, selenite and selenate, found in some plant foods and supplements, can be converted to organic forms in the body. Se-enriched yeast serves as a bioavailable supplement. Additionally, there have also been recent studies on the discovery and bioactivity of Se-containing sugars or polysaccharides in food, which were found to affect diabetes [23,24,25,26], tumor development [27,28,29,30,31], DNA damage [32], as well as immunomodulation [33], intestinal barrier enhancement and gut microbiota regulation [34], neuroprotection [35], and heavy metal detoxification [36]. These organic Se species play a vital role in antioxidant defense, thyroid function regulation, immune system support, and potentially contribute to cancer prevention. And there are synergistic effects of Se with other biological compounds, including vitamins, glucoraphanin, etc. An administration of a combination of Se with vitamin C and/or vitamin D could help boost the immune system via prompting the activity, proliferation, and differentiation of CD4^+^ T cells and increasing the levels of antibodies of immunoglobulin M (IgM), IgG1, IgG2a, IgG2b, and IgG3 [9]. Additionally, it has been suggested that diets supplemented with vitamin C and/or vitamin E, combined with Se, improve antioxidant capacity, immune response, and growth performance in juvenile rainbow trout exposed to ammonia stress [37]. The application of Se can not only promote the accumulation of glucoraphanin and glucosinolates with antioxidant properties [38], but the combination of Se and glucoraphanin can also effectively protect against inflammation and colon tumors. The synergism was confirmed with Se combined with glucoraphanin, promoting the expressions of colonic antioxidant enzymes and phase II enzymes in growing rats [39].

However, careful consideration of dietary intake is crucial to avoid toxicity, as excessive Se can have adverse health. Like many essential elements, Se can be toxic at high doses. The toxicity of organic Se species depends on factors such as the chemical form (derivative), dosage, and individual susceptibility. For example, although the toxicity of SeMet is generally lower compared to inorganic selenium compounds, chronic exposure to high levels of SeMet can lead to selenosis, which may cause symptoms such as gastrointestinal disturbances, hair and nail loss, and garlic breath odor. In a randomized controlled trial, subjects were randomly allocated to receive treatment involving 100, 200, or 300 µg of Se per day in the form of Se-enriched yeast or a placebo yeast. This allocation occurred over a 5-year period from the time of randomization in 1998–1999. Subsequently, the participants were monitored for mortality over an additional 10-year period. The results recommended that high-dose supplements of Se should be avoided due to a higher hazard ratio for all-cause mortality of 300 µg Se/day compared to placebo over the whole follow-up period [40]. A comprehensive systematic review and dose-response meta-analysis were conducted to investigate the link between environmental Se exposure and the risk of type 2 diabetes in non-experimental studies. The preponderance of evidence suggests a direct correlation between blood, dietary, and urinary Se levels and the risk of diabetes. The observed association exhibited a non-linear pattern, with the risk increasing beyond a daily dietary Se intake of 80 μg. Notably, blood Se concentration at 160 μg/L was associated with a risk ratio of 1.96 compared to a concentration of 90 μg/L, which corresponds to an approximate daily Se intake of 60 μg [41]. In addition, a high dose of Se application can also result in Se stress that inhibits the growth of plants. Maize grown in soil spiked with a concentration of 20 mg kg^−1^ sodium selenate (Na_2_SeO_4_), compared to groups with different concentrations (i.e., 0. 2.5, 5.0, and 10.0 mg kg^−1^) of sodium selenate applied to the soil, had a considerable reduction in dry matter, root length, antioxidant enzymes, and other physiological parameters [42]. Our existing knowledge regarding Se content, distribution, bioconversion, and speciation in various agricultural and food products remains incomplete. Furthermore, our understanding of the health benefits and potential toxicities of organic Se compounds present in naturally occurring Se-containing products is limited. It is not sufficient to base dietary recommendations solely on the total Se content of food, as different Se species possess distinct physicochemical properties and biological effects. Based on the metabolite of Se, SePs potentially contribute to the overall nutritional value of Se-containing foods, so studying SePs in food is critical for assessing their nutritional value, understanding their roles in human health, and ensuring that dietary recommendations are based on sound scientific evidence. Consequently, there is a growing emphasis on improving our understanding of the cultivation and preparation methods for Se-biofortified foods, as well as the organic Se species present in these foods and their associated health advantages. This review aims to provide an overview of current research findings pertaining to the composition and structural characteristics of SePs, and selenopeptides (SePPs) found in food sources. Additionally, it summarizes the bioavailability, biological activities, and the structure- activity relationships of these Se-containing compounds.

## 2. Metabolic Processing of Se Compounds in Plants and In Vivo

### 2.1. Inorganic Se and Metabolic Processing to Organic Se in Plants

Selenium is accumulated into the plant tissues through its presence in water, soil, and air, and thereby introduced into the food chain. The uptake of inorganic Se from soil by plants is influenced by several factors, which include pH, salt levels and organic contents of the soil, and the Se assimilation characteristics of the plant species [43]. Inorganic Se exists in diverse valence states. The major inorganic form of dietary Se from plants includes selenate and selenite salts (Table 1), such as Na-selenate (NaSeO_4_) and Na-selenite (NaSeO_3_), respectively. The physical and chemical resemblance between Se and sulfur (S) implies that both these elements possibly share common metabolic pathways in plants [44]. It is likely that the sulphate transporters in plants are also selenate transporters used to assimilate Se into the plant. After the uptake of Se from the soil by plant roots, selenate is translocated to shoot tissues, therefore composing the dominant Se compound in xylem sap. Both selenite and selenate can be incorporated as organic selenoamino acids and, eventually as SePs. To elaborate, selenite is metabolized into SeMet and mainly remains in the root. Selenate is reduced to selenite in its biotransformation route to selenoamino acids (Figure 1) and involves the consecutive action of two enzymes: adenosine triphosphate sulfurylase (ATPS) and APS reductase (APR). Both APS and APR are located in both the chloroplast and the cytosol; however, the chloroplast is where selenate reduction is most likely to occur. Firstly, APS couples selenate to adenosine triphosphate (ATP), creating adenosine phosphoselenate (APSe), which is subsequently reduced to selenite by APR. Sulfite reductase (SiR) may also be involved in the reduction of selenite to selenide in the chloroplast, similar to sulfite reduction. In addition, the nonenzymatic reduction of selenite by the reducing glutathione (GSH) has also been postulated to be involved in the process [45]. Under the action of *O*-acetylserine (OAS) thiol lyase (also called cysteine synthase), which likely takes place in cytosol, chloroplasts, and mitochondria, selenide can be coupled to OAS to form SeCys. *O*-acetylserine is a signal molecule generated by serine acetyl transferase, upregulating the activity of sulfate assimilation enzymes and sulfate transporter. The first produced organic Se compound, SeCys, can be incorporated into proteins nonspecifically, in analogy with Cys, and it can be converted to SeMet, or it can be converted to dimethyldiselenide (DMDSe), elemental Se (Se (0)), or methyl-SeCys [46]. In short, the conversion of SeCys to SeMet involves the actions of three enzymes: cysthanthionine-γ-synthase (CγS), cysthathionine-β-lyase (CβL), and Met synthase. First, SeCys couples to O-phosphohomoserine (OPH) to form Se-cystathionine via CγS; subsequently, Se-cystathionine is converted to Se-homocysteine by CβL, and finally, it is converted to SeMet via the action of Met Synthase. The produced SeMet can follow multiple routes. First, it can also be misincorporated into proteins, or it can be methylated to methyl-SeMet under the action of methionine methyltransferase. Further metabolism leads to the production of dimethylselenide (DMSe), which is volatile and provides a way for excess Se to exit plants [47]. In another metabolic pathway of SeCys, the conversion of SeCys to Se (0) is via the action of selenocysteine lyase (SL). Methylated SeCys, methyl-SeCys, is not incorporated into proteins, and thus it can be accumulated in plants safely [48]. Alternatively, it can be further metabolized to another volatile Se compound, DMDSe, the predominant volatile Se form produced by Se hyperaccumulators [44,45].

### 2.2. Metabolic Fates of Dietary Se In Vivo

The metabolic fates of dietary Se (organic and inorganic) in vivo are shown in Figure 2a. Upon absorption, Se compounds enter the mesenteric venous drainage and is then directed to the liver via the hepato-portal vein. Similar to the utilization of inorganic Se in plants, various forms of inorganic and organic Se are metabolized to hydrogen selenide (HSe^−^), which plays the role of a central gateway for both utilization and excretion of Se in the mammalian body, including incorporation into selenoproteins as selenoamino acids, conversion to selenosugars, methylation, and excretion via urine, feces, and respiratory pathways. The uptake of Se from inorganic Se is presumed to be realized via passive diffusion (selenite) and active transportation (selenate) in the small intestine [56]. Both selenite and those forms that are directly absorbed are reduced to selenodiglutathione (GSSeSG) by the thioredoxin system that involves thioredoxin reductase (TrxR) and thioredoxin (Trx), as well as the glutathione peroxidase system that involves glutathione peroxidase (GPx) and GSH. Oxidized GSSeSG is then transformed into hydrogen selenide, which can then be directed to the metabolic pathways leading to the biosynthesis of selenoamino acids and subsequently selenoproteins.

In contrast to inorganic Se, the absorption of organically sourced Se (i.e., SeCys, SeMet, and MeSeCys) is through transcellular pathways mediated by transporters which are shared with their sulfur-containing analogues. Selenomethionine can be metabolized through three different pathways. Firstly, SeMet is converted to SeCys via trans-sulfuration, and then further metabolized to hydrogen selenide via the trans-selenation pathway, in analogy with the trans-sulfuration pathway. Secondly, SeMet is degraded to methylselenol by γ-lyase, which can be further metabolized to selenide or dimethylselenide for exhalation, or trimethylselenonium ion for excretion. Thirdly, SeMet can also be randomly incorporated into proteins as a substitute for Met in the synthesis of SeMet containing selenoproteins, including hemoglobin and serum albumin. Unlike the metabolism of SeMet, it has not been reported that the dietary intake of SeCys can incorporate into proteins non-specifically or go through direct methylation by γ-lyase. In the body metabolism, it can be converted to hydrogen selenide, followed by the mechanism pathways of hydrogen selenide. In humans, MeSeCys from diets is transformed by *β*-lyase to methylselenol and then metabolized to selenide [12,57,58]. In addition, in maintaining a low body Se status due to Se cytotoxicity, excess selenide is converted into selenosugars for excretion in the urine, such as Se-methyl-N-acetyl-glucosamine (MSeGluNAc), Se-methyl-N-acetyl-galactosamine (MSeGalNAc), and Se-methyl-N-amino-galactosamine (MSeGalNH_2_) [51,59].

The selenoprotein synthesis pathway in eukaryotes is shown in Figure 2b, where the serine (Ser) moiety acts as the backbone for SeCys. It is worth noting that unlike most amino acids and the metabolism of SeCys in plants, SeCys is not incorporated into proteins non-specifically and directly by the ribosome during protein synthesis. The process is initiated by charging Ser onto a dedicated tRNA (tRNA^[Ser]Sec^) to form Ser-tRNA^[Ser]Sec^ by seryl-tRNA^[Ser]Sec^ synthetase (SerS), which has identity elements for Ser rather than SeCys. The seryl residue of Ser-tRNA^[Ser]Sec^ is subsequently phosphorylated by phosphoseryl-tRNA kinase (PSTK) and generates Pseryl-tRNA^[Ser]Sec^, which is then converted to SeCys-tRNA^[Ser]Sec^ using monoselenophosphate as a source of Se. The produced SeCys-tRNA^[Ser]Sec^ transfers SeCys into nascent selenoproteins cotranslationally, through a mechanism involving several dedicated cis elements, including the selenocysteine insertion sequence (SECIS) located in the 3’ untranslated region (UTR) of selenoprotein mRNAs. This process also requires protein factors which act in trans, including SECIS-binding protein 2 (SBP2) and the selenocysteine-specific translation elongation factor (EFSec) [60]. Interestingly, it has been shown that SeCys is encoded by UGA, which is one of the stop codons, but is occasionally redefined as a SeCys codon under the interaction of these SECIS elements, and translated to give SeCys [61,62,63]. SECIS forms a stem and loop structure that acts as a SeCys insertion sequence in the mRNA. It will be recognized by EFSec and SBPS, and, combined with them, this structure delivers SeCys-tRNA^[Ser]Sec^ to the correct position for insertion during selenoprotein synthesis [64].

## 3. Selenoproteins

Selenoproteins (SePs) from Se-enriched agricultural foods have attracted increasing attention due to their bioactivities, indicating that Se-containing foods have great potential to be used as natural functional materials for dietary Se supplements. Selenoproteins account for a significant portion of the total Se content in various Se-enriched foods. It can be obtained from plant-based, animal-based sources, and also fungi [65,66,67,68] and yeast sources [5,69], which not only provide essential amino acids but also possess physicochemical properties of both Se and proteins. Additionally, Se-containing peptides (SePPs) have also been prepared from Se-enriched plants, such as rice, green tea, soybean, and tuna to explore their potential health benefits [20,70,71]. In addition, a variety of factors, including components, amino acid species and sequences, molecular weight, Se status, and structure can significantly affect the bioactivities and functional applications. As several of the SePs identified in mammals are critical selenoenzymes in cells, animal foods that are rich in these SePs are of particular importance. Although Se is not deemed as a crucial element for higher plants, some plants can still integrate it into SePs. Several SePs have been identified in higher plants, including those found in mammalian cells such as GPxs, TrxRs, and selenocysteine methyltransferases. These proteins are also involved in various plant physiology processes, such as antioxidant defense, redox regulation, and Se metabolism [43,72].

### 3.1. The Route from Selenoamino Acids to Selenoproteins

Selenoamino acids are present in various Se-containing foods, including soybean [73], rice [18,70], nuts and seeds [74], corns [75], violifolia [76], potato [77], mushroom [78], yeast (e.g., *Saccharomyces cerevisiae*) and some animal products such as seafood and organic meats [79]. However, the level of selenoamino acids in these foods vary widely depending on factors such as the content of Se in the soil where the food is grown and plant species differences, leading to inconsistencies in the SeP composition and content. It has been reported that SeCys, SeMet and MeSeCys are the primary selenoamino acid species found in plants and animals, and they can replace cysteine (Cys) and methionine (Met), respectively, in protein synthesis [80]. These organic Se compounds possesses higher bioavailability than inorganic Se [59]. Additionally, in many previous studies, it was found that a higher portion of protein-bound SeMet is observed in various SePs [74,81,82,83,84]. SeMet is the most common form of Se found in foods, and it can be easily absorbed by the human body, resulting in its high bioavailability [84]. Selenocysteine, on the other hand, is less common in foods and may have lower bioavailability than SeMet. It can be found in certain dietary sources, including soybean [73], corn [75], and rice [75], but its bioavailability can vary depending on the source. As mentioned above, our current understanding of SeP biosynthesis is that SeCys is incorporated into proteins through genetically encoded mechanisms via the normal protein synthesis pathway but utilizes specific selenoamino acids codons. In contrast, SeMet can be incorporated randomly through non-specific methionine substitution, not as catalytic amino acids [85]. MeSeCys follows a different way, as it is first converted to methylselenol by β-lyase. It is primarily excreted in urine and exhalation or feces but may also pass in the selenide pool [86]. γ-glutamyl methyl-selenocysteine, present in allium and brassica vegetables, is firstly changed to MeSeCys and goes through the same metabolic pathways as MeSeCys [87]. Selenium and sulfur are chemical elements in group 16 of the periodic table. When incorporating oxygen, these elements are known as the oxygen family of molecules, sharing similar chemical properties. Therefore, the substitution of Cys or Met with SeCys or SeMet may have a limited effect on protein structure and function. For selenoenzymes, as illustrated in Section 2, SeCys is insert specifically into the active site of the protein through a specific codon (UGA) in mRNA in human body [61,88]. In fact, some plant species have been found to contain multiple copies of the genes that encode selenocysteine tRNA and other components of the selenocysteine incorporation machinery, suggesting that the ability to synthesize selenoproteins may be particularly important for plants [89]. Up to now, several selenoenzymes have been identified that are reliant on Se for their catalytic activity, in which the active center contains Se in the form of SeCys moiety [90]. By contrast, to date, SeMet is not typically found in the catalytic site of selenoenzymes. SeMet can be incorporated into proteins during translation instead of methionine if it is present in the growth medium or added to the culture. However, it is not a natural amino acid for selenoenzymes and is not enzymatically converted to the active form of Se in selenoenzymes. Both Cys and SeCys can form reactive thiol (-SH) groups, which are essential for the catalytic function of selenoenzymes. However, SeCys is generally considered to be more reactive and better suited to certain redox reactions compared to Cys. This is due to its lower pKa value, which allows it to react more readily with the oxidizing target, making it well-suited for certain redox actions. However, this does not necessarily make it better suited for all redox reactions compared to Cys. It is important to note that the choice of which amino acid to use in a particular redox reaction depends on various factors, including the specific chemistry nature of the oxidizing reagent and the condition of the active site of the enzyme. It is also important to note that the specific role of selenocysteine and cysteine in selenoenzymes can vary depending on the individual enzyme and its catalytic mechanism. In some cases, Cys is more suitable than SeCys. This is because unwanted side reactions and oxidative damage may happen due to the higher activity of SeCys compared to Cys. Therefore, cysteine may be a better choice for redox reactions where stability and selectivity are important factors.

### 3.2. Functional Properties of Selenoproteins in the Human Body

Based on the SelenoDB database, 25 SeP encoded by genes have been identified in the human body, including glutathione peroxidase (GPx), thioredoxin reductase (TXNRD), and iodothyronine deiodinase (DIO). The glutathione peroxidase (GPx)/reductase system is a major antioxidant defense system in cells that is critical in maintaining cellular redox balance. The molecular mass of GPx ranges from 76ku to 95ku. It is a water-soluble tetrameric protein widely present in the body, containing four subunits that are the same or very similar, each subunit having one Se atom. Up to now, eight different isoforms of GPx (GPx 1–8) have been identified in humans, and five of them are seleno-isozymes that contain Se, including cytoplasmic GPx (CGPx or GPx1), gastrointestinal specific GPx (GI-GPx or GPx2), plasma GPx (PGPx or GPx3), phospholipid hydroperoxide GPx (PHGPx4 or GPx4), and GPx6. Each of these isoforms has been shown to contain Se, with SeCys as the catalytic amino acid in the enzyme’s active site [91]. Their activity can reflect the level of Se in the body. They are present in different tissues as biological catalysts in the removal of harmful metabolic peroxide products such as hydrogen peroxide, lipid peroxides, and organic peroxides from the cytoplasm, cell membrane, and extracellular space. This process uses GSH as the electron donor to the peroxide (Table 2) [92]. Oxidized GSH is regenerated back to reduced GSH by glutathione reductase, which is not a selenoenzyme, using NADPH as the electron donor. The first type, cytoplasmic GPx (CGPx or GPx1) consists of 4 subunits of the same molecular weight of 22kDa to form a tetramer [93]. Each subunit contains one molecule of SeCys [94], widely present in various tissues in the body, with the liver and red blood cells being the most predominant. Its physiological function is mainly to catalyze the GSH participation in peroxidation reactions, removing peroxide and hydroxyl free radicals produced in the process of cellular respiratory metabolism. This action alleviates the peroxidation of polyunsaturated fatty acids in cell membranes. The second type, gastrointestinal specific GPx2, is a tetramer composed of 4 subunits with a molecular weight of 22 kDa. It is only present in the gastrointestinal tract of rodents, and its function is to protect animals from the damage of ingesting lipid peroxides [95]. The third type, plasma GPx3, shares the same composition as CGPx and is mainly distributed in plasma. Its function is not well understood, but it has been confirmed to be related to the removal of extracellular hydrogen peroxide [96] and participation in GSH transport [97]. The last, phospholipid hydrogen peroxide GPx4, is a monomer with a molecular weight of 20 kDa, containing one molecule of SeCys. It shares the amino acid motif of SeCys, tryptophan, and glutamine with other GPxs [98]. Originally isolated from pig hearts and livers, it is mainly found in the testicles, but is also distributed to a small extent in other tissues. Its biological function is to inhibit membrane phospholipid peroxidation [99].

The thioredoxin peroxidase/reductase system (TrxP/TrxR) is another key antioxidant system in cells, essential for maintaining cellular redox balance. TrxP and TrxR have distinct active sites where their catalytic reactions take place. In TrxP, the active site includes a redox-active disulfide bond formed between two cysteine residues, which is crucial for its function as a thioredoxin peroxidase. In contrast, the active site of TrxR contains a SeCys residue and a flavin adenine dinucleotide (FAD) cofactor, which are important for its role as a thioredoxin reductase. Thioredoxin (Trx) is a ubiquitous small 12kDa peptide that contains a redox-active disulfide bond and acts as a reducing agent for TrxP, catalyzing the transfer of electrons peroxides and other oxidative molecules, thereby inactivating their reactivity. TrxP is not a selenoenzyme, whereas thioredoxin reductase (TrxR) is. It catalyzes the reduction of oxidized Trx back to its reduced form using NADPH as the electron donor. TrxR reduces oxidized Trx by transferring electrons from NADPH to the active site SeCys residue and then to FAD, leading to the formation of a reduced Trx molecule. On the other hand, TrxP reduces hydrogen peroxide and organic hydroperoxides using electrons from Trx, which itself receives electrons from TrxR. Therefore, TrxR and TrxP work together to maintain redox homeostasis within cells. Additionally, the Trx/TrxR system plays important roles in various other cellular processes, including DNA synthesis, protein folding, and cell signaling [100,101].

Iodothyronine deiodinases (DIOs) are selenoenzymes with three isoforms present in different tissues. All three isoforms are selenoenzymes with SeCys and two histidine residues in the catalytic domain of the enzyme [102,103], and a substrate-binding pocket that accommodates the thyroid hormone molecule. The core physiological functions of DIOs are to act as biocatalysts for the regulation of the activity of thyroid hormones. According to Figure 3, the activation of thyroid hormone is achieved by catalyzing the conversion of inactive thyroid hormone thyroxine (T4) to the primary biologically active thyroid hormone triiodothyronine (T3) via outer-ring deiodination of T4 by DIO1 or DIO2. The inactivation of thyroid hormone occurs through the conversion of T4 to an inactive reduced form T3 (rT3) via inner-ring deiodination of T4 by DIO1 or DIO3, as well as the conversion of T3 and rT3 to diiodothyronine (3,3′-T2) by DIO1, DIO3, DIO1, and DIO2, respectively. This process regulates the levels of active thyroid hormone in the body, and the deiodination is facilitated by the SeCys residue in the active site of the DIOs. Due to the fact that thyroid hormone is linked to the activity level of body metabolism, the control of thyroid hormone activity would regulate the metabolism of the Se. Since DIOs are selenoenzymes, Se deficiency manifests as thyroid hormone dysfunction, which has been associated with various thyroid-related disorders [53,104,105,106]. Selenophosphate synthetase 2 (SEPHS2) is an enzyme that plays a critical role in the biosynthesis of SePs. SEPHS2 catalyzes the synthesis of selenophosphate, which is the activated form of Se used in the incorporation of SeCys into SePs. Dysregulation of SEPHS2 expression or activity has been associated with cancer and neurological disorders, linked to the down-regulation of SeP levels [107,108]. The active site of SEPHS2 is a complex of amino acid residues that cooperate to facilitate the catalytic activity of the enzyme. The crystal structure of SEPHS2 has been determined to have a conserved ATP binding site and a selenophosphate binding site that located at the interface of two domains of the enzyme. This site is formed by several amino acid residues that are critical for catalysis, including a Cys residue, which is involved in the SeCys formation. Other residues are vital for the catalytic capacity of SEPHS2, such as lysine and aspartate, which are involved in ATP binding and stabilization of the intermediate state [109,110,111].

Selenoprotein methionine sulfoxide reductase B1 (MsrB_1_), also known as selenoprotein R (SelR), is another selenoprotein that plays a role in maintaining cellular redox balance. The active site of MsrB1 is a complex and dynamic region of the enzyme that plays a crucial role in its catalytic activity. The active site of MsrB1 consists of several key amino acid residues that play a critical role in the reduction of oxidized methionine, including catalytic residue SeCys/Cys 95 and the resolving residue Cys 4, as well as Trp 43, His 80, Phe 82, Asp 83, Arg 93, and Phe 97, all of which assist in the catalytic process. MsrB_1_ functions as a methionine sulfoxide reductase, catalyzing the reduction of methionine sulfoxide to methionine. This reaction is important for repairing oxidative damage to proteins, as oxidation of methionine residues in proteins can lead to loss of protein function and accumulation of damaged proteins. MsrB_1_/SelR is also involved in regulating cellular signaling pathways, particularly those involved in cell survival and inflammation. MsrB_1_/SelR has been shown to modulate the activity of various transcription factors, such as nuclear factor kappa-light-chain-enhancer of activated B cells (NF-κB) and activator protein 1 (AP-1), which are involved in the regulation of immune and inflammatory responses [112]. In addition, MsrB_1_/SelR has been implicated in the regulation of cell proliferation and apoptosis, as well as in the development of various diseases, such as cancer, neurodegenerative disorders, and cardiovascular disease [113].

Selenoprotein P (SelP), which contains ten Se atoms per molecule as SeCys, totaling about 60% of plasma Se [114], acts as a carrier of SeCys to tissues. Human clinical studies have shown that its low level is linked to Alzheimer’s disease, type 2 diabetes, and cardiovascular disease [115,116]. Table 2 lists other selenoproteins and their known functions to date.

**Table 2 molecules-29-00136-t002:** Functional selenoproteins.

Selenoprotein	Name	Specific/Rich in	Main Function(s)	Reference
Glutathione Peroxidase (GPx)	GPx1 (or CGPx)	Nearly all Mammalian Tissues	A family of peroxidases that reduces H_2_O_2_, lipid peroxides and organic peroxides from the cytoplasm, cell membrane, and extracellular space of human cells using glutathione as the e^−^ donor in reducing peroxide-induced oxidative stress.	[92,117]
	GPx2 (or GI-GPx)	Gastrointestinal Epithelial Cells (Cytoplasm, Extracellular)		
	GPx3 (or PGPx)	Plasma/Intestine (Extracellular, Plasma)		
	GPx4 (or PHGPx)	Sperm (Biological Membrane/Cytomembrane (Phospholipid))		
	GPx6	Embryonic Tissue/Epithelial Tissue of Olfactory Organs		
Thioredoxin Reductase (TrxR)	TrxR1	Extracellular Matrix	Reduces thioredoxin, the e^−^ donor for peroxiredoxin reduction of H_2_O_2_ and other peroxides. Using flavin adenine dinucleotide (FAD) as a coenzyme, it catalyzes the reduction of thioredoxin (Trx) by NADPH. It participates in various cellular processes, including DNA synthesis, protein folding, and cell signaling, and is implicated in several diseases, including cancer and neurodegenerative disorders.	[100,101,118]
	TrxR2	Mitochondria		
	TrxR3	Testis		
Iodothyronine Deiodinase (DIO)	DIO1	Liver/Kidney	Regulation of thyroid gland secretion, thyroid hormone metabolism, and neuron health.	[53,119]
	DIO2	Pituitary Gland/Skeletal Muscle/Thyroid/Heart/Fat/CNS		
	DIO3	Brain/Fetal Tissue/Placenta		
Selenophosphate synthetase 2	SEPHS2	Testes/Liver/Kidney/Brain	Catalyzing the synthesis of selenophosphate from selenide and adenosine triphosphate (ATP), it serves as the selenium donor for selenoprotein and helps maintain proper functioning of selenoproteins.	[107,120,121]
Selenoprotein methionine sulfoxide reductase B1	SelR/MsrB_1_	Cell Nucleus/Cytoplasm	Maintains cellular redox balance, repairs oxidative damage to proteins, regulates cellular signaling pathways, and regulates cell proliferation and apoptosis.	[112,113]
15-kDa Selenoprotein	Sep15	Various Tissues and Organs	Is involved in oxidative stress regulation, protein folding, thyroid hormone metabolism, and immune function.	[122]
Selenoprotein H	SelH	Brain/Nervous System	Cell cycle regulation. Regulates the activity of the nuclear kernel oxidative enzyme and exhibits potential in cancer prevention.	[123,124]
Selenoprotein I	SelI	Testes	Phospholipid biosynthesis.	[53]
Selenoprotein K	SelK	Endoplasmic Reticulum Membrane	Regulates oxidative stress and endoplasmic reticulum stress. Immunity, inflammation and calcium ion adjustment. Regulates endoplasmic reticulum homeostasis and protein folding. Protects skeletal muscles from damage and is required for satellite cells-mediated myogenic differentiation.	[125,126,127,128]
Selenoprotein M	SelM	Brain/Heart/Liver/Kidney/Skeletal Muscle	Maintenance of Ca^2+^ ions, protein folding, promotion of hypothalamic leptin signaling, and thioredoxin antioxidant activity; overexpression of Sel M; activates Parkin-mediated mitophagy to reduce mitochondrial apoptosis and remove HFD-damaged mitochondria.	[129,130]
Selenoprotein N	SelN	Skeletal Muscle	Growth and development of muscles and protein folding.	[131]
Selenoprotein O	SelO	Brain/Liver/Kidney/Testes	Regulation of redox reactions.	[132]
Selenoprotein P	SelP	Liver/Plasma	Se carrier. Transportation of Se to brain and other tissues of the body. Protein folding. Prevention of ferroptosis-like cell death and stress-induced nascent granule degradation.	[133,134,135]
Selenoprotein S	SelS	Plasma/Endoplasmic Reticulum/Immune cells	Regulation of inflammation and redox reactions.	[136,137]
Selenoprotein T	SelT	Brain	Regulation of neuronal function and protection against oxidative stress; regulation of a variety of cellular processes, including calcium signaling, endoplasmic reticulum (ER) stress response, and regulation of protein synthesis.	[138]
Selenoprotein V	SelV	Testes	Expression of taste, regulation of redox homeostasis, and protection against oxidative stress.	[53]
Selenoprotein W	SelW	Mitochondria/Skeletal Muscle/Heart/Brain/Liver/Testes	Oxidative stress regulation, bone remolding and muscle growth. Ensures physiological bone remodeling by preventing hyperactivity of osteoclasts.	[124,139,140]

CNS: central nervous system; CGPx: cytoplasm glutathione peroxidase; GI-GPx: gastrointestinal-glutathione peroxidase; PGPx: plasma glutathione peroxidase; PHGPx: phospholipid hydroperoxide glutathione peroxidase; MsrB_1_: methionine sulfoxide reductase B1; SEPHS2: Selenophosphate synthetase 2.

### 3.3. Dietary Sources of Selenoproteins

It has been evidenced that a dietary supplementation of Se can provide many health benefits, such as neuroprotective function, anti-aging effects, hepatic protection, etc. (Table 3). A daily administration of sodium selenite (300 ng/g body weight) has beneficiary roles in ameliorating neuroinflammation induced by lipopolysaccharide (LPS), including reducing oxidative stress, improving blood-brain barrier integrity, suppressing glial activation, shifting microglial MI/M2 polarization, as well as down-regulating pro-inflammatory cytokines in Se-supplemented mouse brain. In addition, the administration of Se can also improve cognition by reducing neural cell death rate. The neuroprotective functions of Se were ascribed to the facilitated expression of SePs, including GPx4 and Sel P [141]. Furthermore, dietary supplementation of sodium selenite was shown to alter the composition of gut microbiota towards a better microbiota health profile. For example, in the study of Huertas-Abril, et al. [142] the ameliorative effect of a low dose (120 μg/kg bodyweight/day) of sodium selenite-supplemented diet recovered liver function after antibiotic administration in mice. This occurred through the homeostasis of bile acids and cholesterol in the liver, which might be mediated by the gut microbiota. All these findings point to the possible use of sodium selenite as a functional supplement to support body health.

However, the safe and effective use of inorganic Se, such as sodium selenite as a supplement, is still a challenge, considering its low bioavailability and high cytotoxicity. Numerous evidence indicates that organic Se from foods is more effective than inorganic Se in providing antioxidant protection to cells and tissues, thereby contributing to the many documented health benefits of inorganic Se [18,19,20,21,22]. Selenium also occurs naturally as SeP in many plants and fungi, including *Cardamine violifolia* [76,143], soybean [71,73,84,144,145,146], corn [75], brown rice [18,70,147,148], rice [81,83,149,150], algae [151,152,153,154,155], mushroom [78,156,157], peanut [82], maize, cowpea, groundnut [158], etc. As different food categories contain a variety of inorganic and organic Se compounds, their Se profiles vary markedly. Therefore, it is necessary to delineate Se speciation in foods in order to understand their bioavailability and impact on health.

**Table 3 molecules-29-00136-t003:** Food source of selenoproteins and correlated biological effects.

Source	Se Content (μg/g)	Major Se Species	Identification Method	Study Model	Biological Effects	Reference
Se-enriched *Cardamine violifolia*	2450 ± 80	SeCys, SeMet, MeSeCys	HPLC-AFS LC-MS/MS	In vitro: antioxidant activity assessment assays (DPPH/OH/O_2_^−^·scavenging capacity test) In vivo: ICR mice (male, four-week-old, SPF grade).	Antioxidant activity and anti-fatigue activity (increase in SOD level, GSH level, and HG level, promotes GPxs activity, suppress MDA and protein carbonyl levels, decrease in BLA and BUN levels).	[76]
Se-enriched *Cardamine violifolia*	215–735	SeMet, MeSeCys, SeCys	HPLC-AFS Nano LC-MS/MS Preparative HPLC	In vitro: antioxidant activity assessment assays (DPPH/OH/O_2_^−^/ABTS+·scavenging capacity test).	Antioxidant activity.	[143]
Se-enriched soybean	0.33925	SeMet, SeCys	AFS	In vitro: Caco-2 cell In vivo: BALB/c mice (female, six to eight-week-old,18.0 ± 2.0 g body weight).	Antioxidant activity: the presence of SeP from soybean inhibited oxidative stress through upregulating the expression of antioxidant enzymes (GPx, SOD) via modulating the NRF-2/HO-1 signaling pathway. Additionally, the administration of soybean SeP to mice improved the activity of GPx and SOD.	[71]
Se-enriched soybean	6.35–11.47	SeMe, SeCys, SeMeCys	AFS, HPLC ICP-MS, FT-IR SEM	/	/	[84]
Se-enriched soybean	~40	SeMet, SeCys	AFS Q Exactive Orbitrap MS HPLC-MS/MS	In vitro: Caco-2, HepG2 and Endothelial EA. Hy926 cells. In vivo: ICR mice (female, six-week-old) (_D_-galactose-induced aging mice).	Antioxidant activity: protected cells by suppressing the form of TNF-α inflammatory factors and down-regulating the expression levels of cellular adhesion factors. Anti-inflammation and anti-aging: the administration of SeP enhanced SOD and GPx-1, reduced aspartate aminotransferase, amine aminotransferase, and NF-κB, and alleviated brain oxidative damage via modulating MAPK/NF-κB pathway in _D_-galactose-induced aging in mice.	[146]
Se-enriched soybean	Soybean protein isolate: 13.79 ± 0.11 Soybean peptides: 21.78 ± 0.17	SeCys	HPLC-ESI-MS/MS	In vivo: Sprague Dawley rats (male).	Hepatoprotective effects (alleviated liver fibrosis caused by CCL_4_ by promoting GPxs synthesis and increasing MMP9 mRNA expression).	[73]
Se-enriched soybean	1.118	Se-MeSeCys, SeMet	AAS MRM HPLC-ESI-MS/MS	*/*	/	[145]
Se-enriched soybean	75 ± 5	SeMet, SeCys	ICP-MS 2D HPLC-ICP-MS; HPLC-Chip-ESI-ITMS	*/*	/	[144]
Se-biofortified corn (*Zea mays Lin*)	32.37	SeCys, SeMet, MeSeCys	AFS HPLC-ESI-MS/MS	In vitro: antioxidant activity assessment assays (DPPH/OH/O_2_^−^·scavenging capacity test, inhibition of linoleic acid peroxidation) In vivo: BALB/c mice (male, SPF grade)	Antioxidant, hepatoprotective (suppressed MDA, improved SOD and GPxs activities, decreased oxidative stress, inhibited hepatic injury).	[75]
Se-enriched rice	Water-soluble SeP: 22.01 ± 0.34; alkali-soluble SeP: 8.26 ± 0.40; salt-soluble SeP: 1.67 ± 0.07; alcohol-soluble SeP: 0.073 ± 0.13	/	AFS	In vitro: antioxidant activity assessment assays (DPPH/OH) In vivo: Kunming mice (male, four-week-old, 20–25 g body weight)	Antioxidant activity: high free radical (DPPH, OH) scavenging effect. The administration of rice SeP (25 μg/kg/day) enhanced the activities of T-AOC, GPx, SOD, reduced MDA levels, reduces adipocytes, alleviates body weight, liver damage, and the abnormal decrease of the liver coefficient in aging mice. However, the high dose of SeP administration was found to cause hypertoxicity.	[149]
Se-enriched rice	12.84 ± 0.05	SeMet	ICP-MS, RP-UPLC-Triple-TOF MS/MS	In vitro: RAW264.7 cell study	Immunomodulatory activity: the SeP hydrolysate enhanced phagocytosis and proliferation of RAW 264.7 cell and suppressed NO production. However, phagocytosis rate declined when the SeP hydrolysate concentration exceeded 100 μg/mL.	[83]
Se-enriched brown rice	SeP hydro lysates: 0.156–1.79	SeMet	AFS, Scide Triple TOF-LC-MS/MS	In vitro: RAW264.7 cell study	Anti-inflammatory: suppressed the production of NO, PGE_2_, IL-6, IL-1β and TNF-α; inhibited the expression of iNOS and COX-2.	[18]
Se-enriched rice	/	SeMet	SEC-HPLC HPLC-ICP-MS	In vitro: PC12 cell and RAW264.7 cell study	Protected against Pb_2_^+^ induced apoptosis.	[81]
Se-enriched brown rice	6.26	SeCys, MeSeCys, SeMet	2D-LC, HPLC-ICP-MS, ESI FT-ICR MS	In vitro: antioxidant activity determination (DPPH/ABTS + scavenging capacity test, ORAC value, chromium VI-reducing activity, and inhibition activity of linoleic acid emulsion peroxidation)	Antioxidant activity: SeP isolated from brown rice possessed higher ORAC values and free radical scavenging activity than native protein.	[70,148]
Se-fertilized maize, cowpea and groundnut	/	SeMet, SeMeSeCys, SeCys	ICP-MS, HPLC-ICP-MS	/	/	[158]
Se-enriched peanut	9.71	SeMet, SeCys, MeSeCys	ICP-MS, HPLC-ICP-MS	In vitro: AML-12 cell. In vivo: ICR mice (four-week-old)	Exhibited antioxidant activity: peanut SeP suppressed oxidative stress, reversed cell viability and cell death, inhibited ethanol-induced cytochrome P4502E1 activation, and restored GPx enzyme levels. Ameliorated alcohol-induced liver damage: the administration of peanut SeP reduced oxidative stress through modulating MAPK/NF-κB pathway, regulated lipid metabolism, and minimized liver damage.	[82]
Se-containing *Spirulina platensis*	0.67–1.99		ICP-MS	In vitro: antioxidant assessment assay (ABTS+); RAW264.7 cell study	Exhibited antioxidant and anti-inflammatory activities (suppressed inflammatory cytokines, including IL-6, TNF-α, MDA, and IL-1β; decreased the production of NO but promoted the activities of SOD and GPxs).	[151]
Se-containing *Spirulina platensis*	/	/	ICP-AES	In vitro: MC3T3-E1 mouse preosteoblast cells	Prevented mitochondrial dysfunction: balanced the expression of the Bcl-2 family while controlling the opening of the mitochondrial permeability transition pore (MPTP). Additionally, recovered oxidative damage induced by cisplatin. This effect was achieved by inhibiting the excessive generation of reactive oxygen species (ROS) and superoxide anions. Consequently, the process reversed both early and late apoptosis triggered by cisplatin, as it inhibited the cleavage of PARP and the activation of caspases.	[152]
Se-enriched *Chlorella vulgaris*	/	SeMet, SeCys, MeSeCys	ICP-MS, GC-APCI-HRMS, HPLC-ICP-MS GC, MS	*/*	The SeP from *Chlorella vulgaris* has higher bioaccessibility (∼49%) as compared to Se-supplements (∼32%), Se-yeast (∼21%), and Se-foods.	[155]
Se-containing monkeypot nut seeds (*Lecythis minor*)	4480 ± 22	SeMet	ICP-MS, ESI-Q-TOF LC–MS/MS	*/*	/	[74]
Se-enriched mushroom (*Agaricus blazei*)	8.2–26.1	SeCys, MeSeCys, SeMet	HG-AFS HPLC-MS/MS	*/*	/	[157]
Se-enriched mushroom (*Agaricus bisporus*)		SeCys	LC-ESI-MS	In vivo: Sprague Dawley rats (male, 9-week-old).	Antioxidant activities and protection against colorectal cancer (promoted the gene expression of GPx-1 and GPx-2 and enzyme activity of GPx-1 in rat colon).	[78,156]

HPLC: high performance liquid chromatography; RP-UPLC: reversed phase ultra-performance liquid chromatography; AAS: atomic absorption spectrometer; AES: atomic emission spectroscopy; AFS: atomic fluorescence spectrometry HG-AFS; ICP: inductively coupled plasma; MS: mass spectrophotometer; SEC: size exclusion chromatograph; ESI: electrospray ionization; FT-ICR MS: Fourier transform ion cyclotron resonance mass spectrometry; TOF-MS: time-of-flight mass spectrometry; Q-TOF-MS: Quadrupole time-of-flight mass spectrometry; GC-APCI-HRMS: gas chromatography atmospheric pressure chemical ionization high resolution mass spectrometry; MRM: multiple reaction monitoring; FR-IR: Fourier-transform infrared spectroscopy; SEM: scanning electron microscope; ABTS: 2,2′-azino-bis-3-ethylbenzthiazoline-6-sulphonic acid scavenging assay; CCL_4_: chemokine ligands 4; MMP9: matrix metallopeptidase 9; IL-6: interleukin 6; TNF-α: tumor necrosis factor-α; MAPK: mitogen-activated protein kinase; NF-κB: Nuclear factor kappa-light-chain-enhancer of activated B cells; T-AOC: total antioxidant capacity; MDA: malondialdehyde; IL-1β: interleukin-1β; NO: nitric oxide; SOD: superoxide dismutase; GPxs: glutathione peroxidase; PGE_2_: prostaglandin E_2_; iNOS: inducible nitric oxide synthase; COX-2: cyclooxygenase-2; GSH: glutathione; HG: hepatic glycogen; BLA: blood lactic acid; BUN: blood urea nitrogen; Bcl-2: B-cell lymphoma 2; PARP: poly ADP ribose polymerase; DPPH: 1,1-diphenyl-2-picrylhydrazyl free radical scavenging assays; ORAC: oxygen radical absorbance capacity; MM9: matrix metalloproteinase 9; Caco-2 cell: human colorectal adenocarcinoma cell; RAW 264.7: macrophage cell line that was established from a tumor in a male mouse induced with the Abelson murine leukemia virus; PC12 cell: a cell line derived from a pheochromocytoma of the rat adrenal medulla; HepG2 cell: hepatocellular carcinoma cell line; Endothelial EA. Hy926 cells: human umbilical vein endothelial cell line; AML-12 cell: alpha mouse liver 12 cell; MC3T3-E1 cell: osteoblast precursor cell line derived from Mus musculus calvaria.

#### 3.3.1. Preparation and Characterization of Selenoproteins from Foods

A large number of SePs have been isolated from plants, algae, fungi, and yeast. The composition and Se content of these SePs vary among these sources [158], as they are influenced by the regional areas in which they are cultivated and whether Se biofortification was employed [150,158]. The bioaccessibility of Se in rice biofortified with Se^VI^ (sodium selenate) via soil (40 g/ha, 80 g/ha) or foliar spray (20 g/ha, 80 g/ha), or biofortified with Se^IV^ (sodium selenite) via foliar spray (20 g/ha, 80 g/ha) was investigated. It was found that the application of inorganic Se effectively increased the bioaccessible fraction of SeP in crop, and the biofortification through foliar spray was more effective in accumulating Se in bioaccessible protein fractions than soil applications, with up to 2316 μg/kg (foliar spray) and 783 μg/kg (soil application), respectively. Moreover, the total Se content was higher in the rice fortified with sodium selenite via foliar spray than the one fortified with sodium selenate via foliar spray, which means that selenite is more effective in Se biofortification for rice [150]. Similarly, the study of Muleya, Young, Reina, Ligowe, Broadley, Joy, Chopera and Bailey [158] observed that biofortification with potassium selenate (20 g/ha, 75 days) obtained grains from the legumes and maize with higher Se concentration (123–836 µg/kg) than the soil-derived grains (10.7–30.7 µg/kg). Both biofortified and soil-derived Se were transformed into similar Se species, with more than 90% as organic forms and as SeMet in maize (92.0%), groundnut (85.2%), and cowpea (63.7%) from biofortified crops. The mean bioaccessibility of the Se from the biofortified grains was 73.9%, with no significant difference across all crops, but there was a higher bioaccessibility of Se in the grains of legumes than in maize. Moreover, Se-enriched yeast is another way to obtain organic Se species, including SePs and selenoamino acids. Se-enriched yeast can be obtained through growing yeast in Se-containing cultures. Commonly, a culture with 30 μg/mL Na_2_SeO_3_ can result in Se accumulation in the range of 1200–1400 μg/g dried yeast (*Saccharomyces cerevisiae*), and the inorganic Se can be bio-transformed in yeast to form SePs [159,160].

The selenoprotein extraction method can influence extraction efficiency, SeP purity, and Se content. In a study of SeP from Se-enriched rice, ultrasound-assisted alkaline (0.09 M NaOH) extraction (UA) obtained 12.84 µg/g, which was a much higher level compared to ultrasound-assisted enzyme (α-amylase) extraction (UE) at 4.26 µg/g [83]. Moreover, the purification and extraction yields were significantly different, with higher purity (75.61%) but lower yield (77.83%) of SeP with the UA method compared to 72.21% and 84.42%, respectively, with the UE method. Water, salt (0.5 M NaCl), alcohol (75% ethanol), and alkali (0.1 M NaOH) extraction of SeP from Se-enriched rice revealed that the protein and Se contents in these different extraction solvents were different and were in an order of water-soluble proteins > alkali-soluble proteins > salt-soluble proteins > ethanol-soluble proteins [126].

SeMet it the predominant chemical form of Se in SePs from Se-enriched food, including soybean [84], rice [83], peanut [82], maize, cowpea and groundnut [158]. This is due to the fact that while SeMet, SeCys, and MeSeCys are common selenoamino acids, SeCys can be further transformed to selenocystathionine, selenohomocysteine, and finally to SeMet (Figure 1). However, recent observations suggest that selenoamino acids other than SeMet are dominant in some foods. Se-biofortified soybean, for example, contains mainly SeCys in its SeP [73,146]. In Se-enriched mushrooms, the predominant Se compound was SeCys, found as free selenoamino acids or within the SePs from *Agaricus bisporus* [78] or *Agaricus blazei* [157]. In Se-enriched soybean, Se-MeSeCys and SeMet exist in the soybean SePs but with Se-MeSeCys contributing 66.4% of the total Se [145]. Similarly, SeCys, Se-MeSeCys, and SeMet are the main organic Se forms in Se-enriched brown rice, which accounted for 44.3% of the total Se content [148].

Low MW SePPs have gained increasing attention owing to their higher bioavailability, absorbability, and bioactivity compared to higher MW SePPs and SePs. For example, low MW SePPs from the plant *Cardamine violifolia* showed higher free radicals scavenging activity than the higher MW SePPs [76,143]. Similarly, Se-enriched brown rice protein hydrolysates peptide fraction at 1.0–3.5 kDa possessed more effective anti-inflammatory activity than the fraction with a higher MW [18]. Additionally, SePPs with low-MW (229.4–534.9 Da) from Se-enriched rice significantly protected cells from Pb^2+^ induced apoptosis through a caspase-dependent mitochondrial pathway [81].

Structural analysis indicates that the incorporation of Se from selenoamino acids into protein might affect the protein’s secondary structure, including the α-helix, β-sheet, β-turn, and random coil structures [161]. The resulting change in secondary structure arising from the change in the primary structure might further influence some physicochemical and biological properties of the protein. Within a polypeptide chain, Se can be incorporated as a SeCys residue which can form covalent Se-Se bonds with neighboring amino acids within the protein [137].

SeCys in SePs can influence both the disulfide bond and the secondary structure of proteins, and possibly protein folding, thus altering the protein functional properties. In addition, the tertiary structure of proteins depends on the disulfide bridge formation, which happens during the oxidation of two neighboring Cys. Proteins with diselenide bonds are more likely to undergo reduction than those with disulfide bonds due to the longer length of the diselenide bridge compared to the disulfide bridge, giving it a lower redox potential. But the effects of Se incorporation on protein structure and on the protein functional properties in foods are poorly documented [63].

#### 3.3.2. Biological Activity of Selenoproteins from Food

Food sources of SePs and SePPs and their correlated biological effects are tabulated in Table 3.

It has been observed that SeP possesses higher antioxidant activity than Se-free protein/peptides [70,71,146]. The Se-enriched soybean protein isolate displayed stronger free radical scavenging ability compared to the native soybean protein isolate [71]. Selenoprotein from soybean inhibited induced oxidative stress in Caco-2 cell through upregulating the expression of selenium (GPx) and non-selenium (SOD) antioxidant enzymes and regulating the NRF-2/HO-1 signaling pathway. This pathway is responsible for modulating calcium levels, preventing pyroptosis, ferroptosis, autophagy, alkaliptosis, clockophagy, and programmed cell necrosis [162]. Administration of soybean SeP (5, 20, 40 g/kg body weight/day) to mice improved the activity of GPx and SOD in tissues [71]. A SePP fraction isolated from Se-enriched brown rice has higher oxygen radical antioxidant capacity (ORAC), free radical scavenging activity, and chromium VI-reducing activity compared to original brown rice peptides without Se content [70]. It was also found that the administration of SeP extracted from *Cardamine violifolia* in mice (5, 10, 20 µg/kg body weight/day) increased the levels of GPxs, SOD, and glutathione (GSH) in tissues, while decreasing the levels of malondialdehyde (MDA) (indicative of lipid oxidation) and protein carbonyl (indicative of protein oxidation) in blood [76]. Moreover, dietary supplementation of SeP-containing Se-enriched yeast enhanced both the antioxidant capacity and immune response in juvenile *Eriocheir Sinensis* under nitrite stress [163]. It is recommended to incorporate dietary Se at a concentration of 3.98 mg Se//kg of diet, with 3 mg of Se provided in the form of Se-enriched yeast. This supplementation enhances the growth performance, feed utilization, and positively influence liver and kidney histology in juvenile meagre fish, thereby resulting in potential economic benefits [164].

Selenoprotein isolated from some foods has been shown to possess anti-inflammatory effects. The anti-inflammatory activity of SeP isolated from algae (*Spirulina platensis*) was evaluated on RAW264.7 macrophages. The results showed that treatment with SeP (0.31–125 µg/mL) suppressed production of inflammatory cytokines, including interleukin 6 (IL-6), tumor necrosis factor-α (TNF-α), MDA, and interleukin-1β (IL-1β). Moreover, it led to a decrease in the production of nitric oxide (NO) while increasing the activities of SOD and GPxs [151]. Similarly, the SeP obtained from Se-enriched brown rice was found to suppress the production of NO, prostaglandin E2 (PGE_2_), IL-6, IL-1β and TNF-α, as well as inhibit the expression of inducible nitric oxide synthase (iNOS) and cyclooxygenase-2 (COX-2) in cultured macrophages [18]. Selenoproteins from Se-enriched soybean protected endothelial cells through suppressing the production of TNF-α inflammatory factors and down-regulating the expression of cellular adhesion factors [123]. Administration of SeP (30 µg Se/kg body weight/day) enhanced SOD and GPx-1, reduced aspartate aminotransferase, amine aminotransferase, and NF-κB, and alleviated brain oxidative damage via modulating mitogen-activated protein kinase (MAPK)/NF-κB pathway in _D_-galactose-induced aging mice. Rice SeP hydrolysate applied at 20–100 μg/mL enhanced phagocytosis and proliferation of macrophages and suppressed NO production by the cells.

Selenoproteins also possess hepatoprotective properties [73,75,82,149]. The SeP extracted from soybean alleviated liver fibrosis caused by chemokine ligands 4 (CCL_4_) by promoting GPxs synthesis and increasing the mRNA expression of matrix metallopeptidase 9 (MMP9) in rats [73]. The administration of peanut SeP reduced oxidative stress through modulating MAPK/NF-κB pathway, regulate lipid metabolism, and alleviated liver damage in mice [82]. Administration of rice SeP (10, 25 μg/kg/day) enhanced the antioxidant capacity (T-AOC) and the activities of total GPx and SOD, reduced MDA and adipocytes levels, and alleviated body weight, liver damage and the abnormal decrease of the liver coefficient in aging mice. Importantly, the high dose of SeP administration (50 μg/kg/day) was found to cause hypertoxicity [149].

Other bioeffects of SeP have also been reported. Selenoprotein isolated from *Spirulina platensis* was found to prevent mitochondrial dysfunction [129]. The presence of SeP balanced the expression of the Bcl-2 family while controlling the opening of the mitochondrial permeability transition pore (MPTP) and recovered oxidative damage induced by cisplatin. This effect is achieved by inhibiting the excessive generation of reactive oxygen species (ROS) such as superoxide anions. Consequently, the process reversed both early and late apoptosis triggered by cisplatin, as it inhibited the cleavage of poly ADP ribose polymerase (PARP) and the activation of caspases. Additionally, it was found that the administration of Se-enriched yeast exhibits protective effects against Cd-induced necroptosis injury by mitigating oxidative stress and suppressing the MAPK pathway in the chicken liver [165]. More studies are still required to uncover the bioactivity or alleviative effects of SeP associated with many other diseases.

## 4. Conclusions and Perspective

This comprehensive review documented our knowledge of selenoproteins in plants and fungi, which are a vast source of organic Se from foods. Emphasis was placed on the bioavailability and bioactivities of these food-derived organic selenoproteins, as well as their associated selenopeptides and Se-containing amino acids. These compounds were explored for their antioxidant and anti-inflammatory effects, along with their hepatoprotective functional molecules. In addition, attention was given to their composition, structural features, and structure-activity relationships. Further research in this area might lead to the development of new functional products for use in promoting human health and disease prevention. Future studies on selenoprotein should continue to focus on the identification and characterization of selenoproteins in various foods, assessing their bioaccessibility and bioavailability, investigating the metabolic pathways through which they are processed in the body, and exploring the correlation between these organic selenium forms and their physiological effects within the human body.

## Figures and Tables

**Figure 1 molecules-29-00136-f001:**
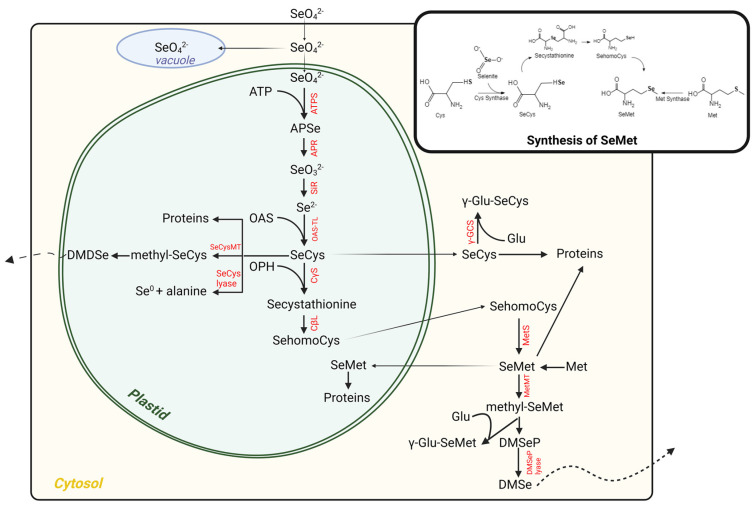
Schematic overview of Se metabolism in plants leading to the production of organic Se. APR: adenosine phosphosulfate reductase; APSe: adenosine phosphoselenate; ATPS: adenosine triphosphate sulfurylase; CβL: cysthathionine-β-lyase; CγS: cysthathionine−γ−synthase; DMDSe: dimethyldiselenide; DMSe: dimethylselenide; DMSeP: dimethylselenopropionate; γ−GCS: γ−glutamylcysteine synthetase; γ−Glu-SeCys: γ−glutamyl-SeCys; γ−Glu-SeMet: γ−glutamyl-SeMet; Glu: glutamate; MetMT: methionine methyltransferase; MetS: methionine synthase; OAS: O-acetylserine; OAS−TL: OAS thiol lyase; OPH: O−phosphohomoserine; SeCys: selenocysteine; SeMet: selenomethionine; SiR: sulfite reductase (or glutathione); SeCysMT: SeCys methyltransferase.

**Figure 2 molecules-29-00136-f002:**
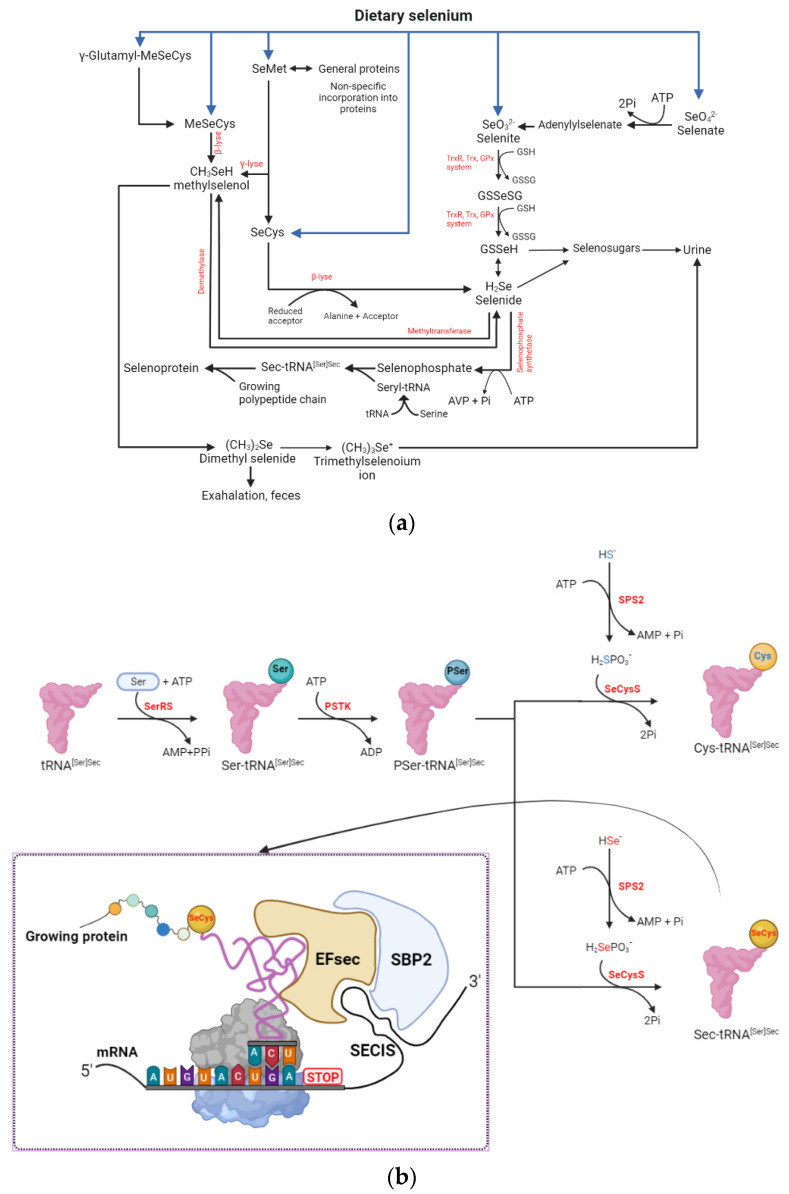
Metabolism of dietary selenium and selenoprotein biosynthesis pathway in the body. (**a**) Metabolism of dietary selenium in the body. (**b**) Mechanism of SeCys biosynthesis in eukaryotes and the SeCys machinery-based pathway for synthesis of Cys. This includes the delivery of tRNA with SeCys to an internal UGA stop codon. ADP: adenosine diphosphate; AMP: adenosine monophosphate; ATP: adenosine triphosphate; EFsec: selenocysteine-specific translation elongation factor; Cys: cysteine; GSH: reduced glutathione; GSSG: glutathione disulfide; GSSeH: glutathioselenol; GSSeSG: selenodiglutathione; MeSeCys: methylselenocysteine or Se-methylselenocysteine; Pi: phosphate; PPi: pyrophosphate; PSTK: phosphoseryl-tRNA kinase; GPx: glutathione peroxidase; SBP2: SECIS-binding protein 2; SECIS: selenocysteine insertion sequence; SeCys: selenocysteine; Sec: SeCys; SeCysS: SeCys synthase; SeMet: selenomethionine; SEP2: SECIS-binding protein 2; SPS2: selenophosphate synthetase 2; Ser: serine; SerRS: seryl-tRNA synthetase; Sec-tRNA^[Ser]Sec^: selenocysteyl-tRNA^[Ser]Sec^; Ser-tRNA^[Ser]Sec^: seryl-tRNA^[Ser]Sec^; PSer-tRNA^[Ser]Sec^: phosphoseryl-tRNA^[Ser]Sec^; TrxR: thioredoxin reductase; Trx: thioredoxin.

**Figure 3 molecules-29-00136-f003:**
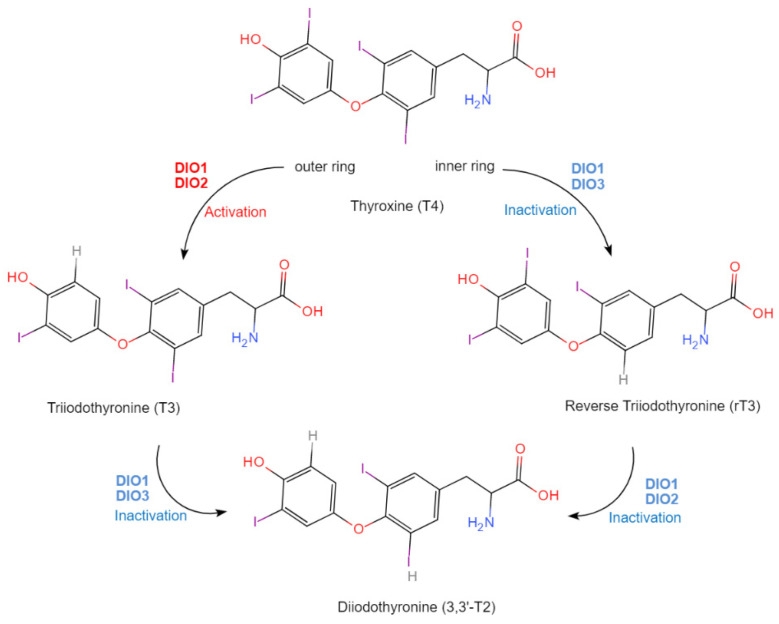
Schematic overview of deiodinase isoforms reactions. DIO1: deiodinase 1; DIO2: deiodinase 2; DIO3: deiodinase 3.

**Table 1 molecules-29-00136-t001:** Inorganic selenium compounds.

Name	Molecular Formula	Chemical Structure	Distribution	Reference
Selenate (SeVI)	SeO_4_^2−^	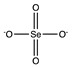	Soil, plants, mammals	[49,50,51,52,53]
Selenite (SeIV)	SeO_3_^2−^	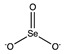	Soil, plants, mammals	[52,53,54,55]
Selenide	Se^2−^	Se^2−^	Plants, mammals	[49,53,55]
Monomethylselenonium	CH_5_Se^+^	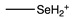	Mammals	[51,53]
Dimethylselenide	(CH_3_)_2_Se	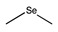	Plants, mammals	[49,51,55]
Trimethylselenonium	(CH_3_)_3_Se^+^	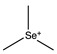	Plants, mammals	[49,51]
Selenophosphate	PO_3_Se^3−^	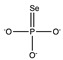	Mammals	[49,53]

## Data Availability

Data will be made available on request.

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
