# Peer review of "Selenoproteins in Health"

_molecules, 2023, doi:10.3390/molecules29010136_

Round 1

Reviewer 1 Report

Comments and Suggestions for Authors

I think this manuscript is a comprehensive review on organic seleno-derivatives in foods and their beneficial effects on health.

It is interesting to read this overview of current research focused on the composition and structural characteristics of SeProteins, and selenopeptides (SePPs) found in food. Additionally, the bioavailability, biological activities, and structure and activity relationship information are reported and are important to the overall review.

However, since the authors state that there is a growing interest in Se-biofortified foods and  the research in this field is still progressing, a paragraph related to the organic Se species toxicity should be added to give attention to the doses, type of derivatives and risks associated.

So the authors should not just highlight the positive aspects, but also underline the potential risks associated to these derivatives. 

Reviewer 2 Report

Comments and Suggestions for Authors

Dear editor, dear authors,

The presented manuscript treats overall the subject of Selenium benefits for human health and emphasis related topics, starting with inorganic and organic sources, with many data about geographic area, soils rich or poor in selenium, as well with details about daily intake in various populations. Furthermore, the authors report data concerning Se metabolism in plants leading to the production of organic Se, in the context which the absorption of organically source is better in contrast to inorganic Se. Information on selenoprotein synthesis pathway, bioavailability, bioaccessibility and bioactivity of various Se forms are given, with careful and rigorous choice of references. In the end, missing and insufficient knowledge regarding the correlation between different organic selenium forms and their physiological effects within the human body, is recognized.  

As recommendation, a point that worth to be addressed, is the synergy of Se with glucoraphanin, vitamins or other trace minerals to attaint benefit for the immune system.

Another aspect to be mentioned is how deficiencies of the daily dose and excessive uptake of Selenium are reflected on health and physiological condition.

Se-enriched yeast worth also to be cited (preparation and characterization of selenoproteins from yeast).
